# *Bacillus subtilis*-Based Probiotic Improves Skeletal Health and Immunity in Broiler Chickens Exposed to Heat Stress

**DOI:** 10.3390/ani11061494

**Published:** 2021-05-21

**Authors:** Sha Jiang, Fei-Fei Yan, Jia-Ying Hu, Ahmed Mohammed, Heng-Wei Cheng

**Affiliations:** 1Joint International Research Laboratory of Animal Health and Animal Food Safety, College of Veterinary Medicine, Southwest University, Chongqing 400715, China; jangsha0527@swu.edu.cn; 2Animal Welfare Institute, College of Animal Science and Technology, Zhejiang A&F University, Hangzhou 311300, China; yanff@zafu.edu.cn; 3Department of Animal Science, Purdue University, West Lafayette, IN 47907, USA; hu165@purdue.edu; 4Department of Animal and Poultry Behavior and Management, Faculty of Veterinary Medicine, Assiut University, Assiut 71526, Egypt; ahmed.abd_elhafez@vet.au.edu.eg; 5USDA-Agricultural Research Service, Livestock Behavior Research Unit, West Lafayette, IN 47907, USA

**Keywords:** heat stress, probiotic, gut microbiota, the gut–microbiota–brain axis, skeletal health, welfare, broiler chicken

## Abstract

**Simple Summary:**

High ambient temperature is a major environmental stressor affecting the physiological and behavioral status of animals, increasing stress susceptibility and immunosuppression, and consequently increasing intestinal permeability (leaky gut) and related neuroinflammation. Probiotics, as well as prebiotics and synbiotics, have been used to prevent or decrease stress-associated detrimental effects on physiological and behavioral homeostasis in humans and various animals. The current data indicate that a dietary probiotic supplement, *Bacillus subtilis,* reduces heat stress-induced abnormal behaviors and negative effects on skeletal health in broilers through a variety of cellular responses, regulating the functioning of the microbiota–gut–brain axis and/or microbiota-modulated immunity during bone remodeling under thermoneutral and heat-stressed conditions.

**Abstract:**

The elevation of ambient temperature beyond the thermoneutral zone leads to heat stress, which is a growing health and welfare issue for homeothermic animals aiming to maintain relatively constant reproducibility and survivability. Particularly, global warming over the past decades has resulted in more hot days with more intense, frequent, and long-lasting heat waves, resulting in a global surge in animals suffering from heat stress. Heat stress causes pathophysiological changes in animals, increasing stress sensitivity and immunosuppression, consequently leading to increased intestinal permeability (leaky gut) and related neuroinflammation. Probiotics, as well as prebiotics and synbiotics, have been used to prevent or reduce stress-induced negative effects on physiological and behavioral homeostasis in humans and various animals. The current data indicate dietary supplementation with a *Bacillus subtilis-*based probiotic has similar functions in poultry. This review highlights the recent findings on the effects of the probiotic *Bacillus subtilis* on skeletal health of broiler chickens exposed to heat stress. It provides insights to aid in the development of practical strategies for improving health and performance in poultry.

## 1. Introduction

The gut microbiota (gut flora) plays a critical role in preserving host health by releasing various metabolites to stimulate hosts’ neural, endocrine, and immune systems [1,2,3,4,5,6,7]. However, microbial functions can be changed markedly by multiple factors, such as lifestyle (living environments), nutrition and health condition (pathophysiological disorders), life-stage (age), and medical substances in humans [8,9,10,11,12,13,14,15]. It can be further affected by management issues such as rearing conditions (restriction, crowding, heat, cold environments), breeding programs (selected for a special production indicator), and stress-related factors (transportation and weaning) in farm animals [16,17,18,19]. Impaired gut microbiota (dysbiosis), alterations in the richness and diversity of microbiota, leads to the pathophysiological processes of various diseases in humans and animals [20,21,22,23,24]. The modulation of the gut microbiota by the administration of probiotics, as well as prebiotics and synbiotics, has become a biotherapy strategy for preventing and treating many diseases, from stress-related irritable bowel syndrome to neuropsychiatric disorders [25,26,27]. Probiotics (direct-fed microbials) can restore the ecologic stability of gut microbiota by inhibiting pathogens, promoting the growth of beneficial bacteria, and releasing bioactive and immunomodulatory factors to improve the function of the hypothalamic–pituitary–adrenal axis (one of the major stress response systems) and immunity via the microbiota–gut–brain axis and or the microbiota–gut–immune axis [28,29].

Commercial broiler chickens have been selected based on growth rate and feed efficiency, reaching an average of 6 lbs. live weight by approximately 6 weeks. The breeding program has caused broilers to be very sensitive to stress and the related gut inflammation due to selection-associated metabolic disorders and immature immune system, mostly during the first few weeks of life [30,31]. Sub-therapeutic levels of antibiotics have been used as growth promotors in broiler production to prevent or control the incidence of clinical and subclinical diseases, such as necrotic enteritis caused by *Clostridium perfringens*, and increasing food digestion and nutrient absorption [32]. However, poultry production has been considered one of the main sources of antimicrobial-resistant (AMR) bacteria, and antibiotics have been banned or restricted in many countries globally. Eliminating the use of antibiotics in poultry production has caused considerable consequences, comprising production performance and increasing the incidence of gut infectious diseases and related mortality [33]. The modulation of aberrant gut microbiota into a healthy state has become a critical objective for the poultry industry. In the past decade, numerous pieces of evidence have suggested that probiotics can be used as antibiotic replacements in poultry production [34,35,36]. Generally, probiotics confer a health benefit to chickens due to their positive effects on microbiota composition (restoring microbial homeostasis), mucosal barrier integrity (reducing gut permeability), and immune response (reducing inflammation). *Bacillus subtilis* is one of the most common bacterial species used in commercial probiotic products for farm animal production [37,38], including in poultry [39,40,41].

*Bacillus spp.* Gram-positive rods have a distinct advantage over *Lactobacillus* and *Bifidobacterium* for use as probiotics due to (1) the ability of sporulation to survive environmental stress, preparation conditions, and application processes; (2) their tolerance to low pH, bile salts, and other harsh conditions in the gastric environment; (3) the maintenance of their viability and desirable characteristics within the gastrointestinal tract (GIT); and (4) their ability to form biofilms to release biochemical compounds [42,43,44,45,46,47,48]. The most commonly used commercial *Bacillus* probiotic strains include *B. amyloliquefaciens*, *B. cereus*, *B. clausii*, *B. licheniformis*, and *B. subtilis* (Table 1). It is known that the health and production effects of probiotics are genera-, species-, and strain-specific. *B. subtilis*, as an ideal multifunctional probiotic bacterium, has been gaining popularity in recent years. It has been used as antibiotic replacement to regulate gut flora, improve growth performance, and enhance immunity and gut health in poultry under various rearing environments and infectious immune challenges [39,49,50,51,52].

In recent years, a growing body of evidence suggests that detrimental gut microbiota leads to bone loss in humans and various animals. The use of probiotics has become a potential therapeutic approach for preventing and treating bone damage due to their positive effects on mineral absorption, anti-inflammatory regulation, and the release of neuroactive factors, including tryptophan, the precursor of serotonin, and stress hormones via the microbiota–gut–brain axis and or the gut–bone axis [53,54,55]. However, probiotic effects are species-specific. The effects of probiotic strains of *Lactobacillus* and *Bifidobacterium* on bone health in humans and experimental animals have been comprehensively reviewed recently [56,57]. To the best of our knowledge, an overview of the effects of the probiotic *B*. *subtilis* and other closely related phylogenetic clades on bone health in poultry is still lacking in the literature. Therefore, the objective of this review is to summarize the available literature and our own data on the current state of knowledge about the effects of *B. subtilis-*based probiotics on the skeletal health of broiler chickens in response to heat stress.

## 2. Heat Regulation in Broiler Chickens

Broiler chickens, like other homeothermic animals, can maintain a relative range of body temperature (named the thermoneutral zone, TNZ, or thermoneutrality) by balancing the rate of body heat production and the rate of body heat loss to the ambient environment. The TNZ of chickens is about 18.3–24 °C (65–75 °F) [58], which varies according to chickens’ genetic background, age, weight, and diet [59]. Within the TNZ, chickens maintain internal (body core) temperature and reproducibility with minimal metabolic regulations, whereas when the ambient temperature reaches the upper or lower critical limit, chickens have to reset their thermal points (thermal comfort decision) via multiple pathways to prevent heat stress or cold stress from causing pathophysiological damage.

The body temperature of homeothermic animals including chickens is regulated by the thermoregulatory center of the hypothalamus. Thermo-receptors are located within the hypothalamus to monitor the temperature of the blood as it circulates in the brain and on the skin’s surface to monitor the external temperature and to sense the rate of temperature changes. The hypothalamus receives the sensory inputs from the thermo-receptors and then sends impulses to the effector organs so that appropriate adjustments are made in response to temperature changes. During hot weather, the hypothalamus normally encourages responses that increase body heat loss and/or reduce body heat production by coordinating the autonomic, endocrine, and somatic motor functions. Consequently, these changes lead to life-surviving responses in chickens, such as vasodilation of superficial blood vessels, panting (hyperventilation, i.e., open mouth breathing with an increased respiratory rate), reduced feed intake and related metabolic rate, and wing spreading (rearranging feathers to positions that facilitate heat loss). Hyperthermia occurs when the hypothalamus loses its thermo-sensitivity to high temperature [60,61].

There are two major thermoregulation systems linked to the hypothalamus (specifically, the preoptic area, POA): the hypothalamic–pituitary–adrenal (HPA) and the hypothalamic–pituitary–thyroid (HPT) axes in homeothermic animals including chickens [62,63,64] (Figure 1). The hypothalamus, as the thermoregulatory center, receives and integrates the internal and external signals, and then activates both the TRH (thyrotropin-releasing hormone released from the hypothalamus)-TSH (thyroid-stimulating hormone released from the pituitary)-T4 (thyroxine from released the thyroid) and the CRH (corticotropin-releasing hormone released from the hypothalamus)-ACTH (adrenocorticotropic hormone released from the pituitary)-CORT (corticosterone released from the adrenal glands) neuroendocrine systems to initiate heat stress or cold stress responses [65].

The most abundant hormone produced by the thyroid gland is T4 (thyroxine, approximately 94% of hormones released by the thyroid gland), and T4 must be converted to T3 (3,5,3′-triiodothyronine) for proper thyroid function. T3 is the active form of thyroid hormones, playing an important role in energy metabolism and thermogenesis [66,67]. The gut plays an important role in the conversion process (the liver is another primary organ), approximately 20% of T4 is converted into T3 by gut bacteria via the thyroid–gut axis [68,69]. Heat or cold exposure reduces or enhances the conversion to decrease or increase metabolic heat production, respectively [70]. Chickens experiencing long-term heat stress have low levels of thyroid hormones, leading to a reduced metabolic rate in order to decrease heat production [71,72].

Corticosterone (CORT, the major glucocorticoid hormone in birds) also plays an important role in body heat production by regulating appetite, adipose tissue metabolism, and energy homeostasis [73]. Overexpression of glucocorticoids leads to the accumulation of brown adipose tissue (BAT) and increased lipogenesis of white adipose tissue (WAT), resulting in metabolic disorders [74,75,76]. In addition, glucocorticoids regulate body temperature through inhibiting the function of the HPT axis, i.e., reducing the synthesis and release of TSH and T4, and the conversion rate of T4 to T3 [77,78]. Glucocorticoids also have roles to counter stress by making critical life-sustaining physiological adjustments via redistributing body energy and nutrients between critical and noncritical biological systems for survival. The gastrointestinal tract (GIT) is considered less vital for immediate survival, and reducing its nutrient supplies leads to poor reproduction and immunosuppression [79].

Behavioral change is the major method used by birds including chickens to release excessive body heat to adapt to the surroundings due to their poor cooling system, i.e., their lack of sweat glands and the fact that they are covered by feathers. Birds’ behavioral responses typically used to cope with heat stress include eating less and drinking more, hyperventilation (panting), seeking cooler areas, and wing spreading (to promote cooling by reducing body insulation). Panting is the mostly important method used by birds to dissipate internal heat, which is accomplished via the evaporation of moisture from the buccal cavity and upper respiratory tract [80,81]. Panting in birds releases approximately 540 calories per gram of water lost by the lungs [82]. If hot conditions persist, excessive panting results in birds expelling great amounts of carbon dioxide, leading to metabolic alkalosis, a serious disruption of the acid–base balance, eventually leading to death caused by heat stroke-associated hypoxia [83,84,85].

## 3. Heat Tolerance and Heat Stress in Commercial Broilers

Heat stress occurs when an animal is unable to cool itself to maintain a healthy body temperature. Chickens can adapt to ambient temperatures up to 25 °C (77 °F), whereas temperatures above this level lead to heat stress. Adult chickens exposed to environmental temperatures greater than 37.8 °C (100 °F) experience more heat gain than heat loss, resulting in an increase in body core temperature (the normal range: 40.6–41.7 °C or 105–107 °F) and potentially death [86].

Heat stress is a detrimental environmental stressor affecting poultry meat production globally. Particularly, global warming over the past decades has resulted in more hot days with more frequent intense and long-lasting unexpected heat waves [87]; and the number of days with a heat index of 37.8 °C (100 °F) or above has become more common in recent years. In particular, modern broiler chickens have been selected continuously for high product output, i.e., maximum growth rate and high feed conversion efficiency over a period of time from 6 to 8 weeks [88]. The breeding program greatly affects broiler physiological homeostasis, leading to immature or impaired metabolism, immunity, and antioxidant status as well as susceptibility to inflammation and infection. High environmental temperatures, especially combined with high humidity, impose severe stress on broiler health and welfare due to poor heat tolerance, with a limited ability to regulate heat loss by feathering and without sweat glands. All these pathophysiological changes negatively affect production performance, accelerate morbidity and mortality, and cause heavy economic losses [89,90]. Excessive mortality due to heat stress is commonly seen in commercial broiler flocks. For those broilers which survive high temperatures, economically important production traits such as feed intake, body weight gain, meat quality, musculoskeletal health, and feed efficiency are detrimentally affected [91,92].

Several methods have been used to prevent or reduce heat stress in poultry, such as fasting and/or dietary adjustments (to reduce metabolic heat production and to maintain nutrient intake), genetic selection (breeding heat-resistant chickens), changes in early-age thermal conditions, supplementation of active substances during embryonic development, adequate ventilation (to increase heat loss via air flow), and cooling systems (to increase cold air circulation) [93,94]. However, the results are not stable and are affected by multiple factors. For example, the most commonly used air condition cooling/ventilation systems can be effective in reducing heat stress, but they have several negative factors, such as (1) noise from the operating cooling system, (2) unevenly distributed chickens, (3) maintenance costs, and (4) unexpected system shutdowns.

Antibiotics have been used for growth promotion, prevention, and the treatment of infectious diseases in broiler production, which has played a major role in the development of the poultry industry for more than 70 years in the United States and several other countries [95,96,97]. However, consumers are becoming increasingly concerned about drug residues in meat products, environmental contamination, and drug-resistant bacteria. Humans are exposed to antibiotic-resistant pathogens when consuming contaminated food and water or coming into contact with animals or contaminated environments. Growing evidence suggests that the routine use of antibiotics in food animals, such as broilers, plays a key role in the development and spread of drug-resistant bacteria and is subsequently associated with an increased risk of foodborne infections by antimicrobial-resistant pathogens. Due to food safety concerns and the public demand for antibiotic-free farm animal products, many countries have banned or are going to ban the inclusion of antibiotics in broiler diets as a routine means of growth promotion. In addition, the vast majority of food producers and distributors (such as Tyson, KFC, Perdue Farms, and McDonald’s) have announced that they will stop sourcing chickens raised with the “routine” use of antibiotics. It is critical to develop reliable methods to control heat stress. Probiotics, as a gut health-supporting additive, have been proposed as alternatives to antibiotics in broiler production [35].

## 4. Heat Stress and Skeletal Health of Broilers

The skeleton system (bones) in broilers, like in all vertebral animals, is continuously remodeling in response to internal and external stimuli. As an internal factor, commercial broilers have been selected at their full genetic potential for fast growth, great feed efficiency, and a high percentage of breast muscle. Due to the breeding program, young commercial broilers have the body size of adult native chickens, coupled with an immature skeletal system [98]. Their bones grow very rapidly, with nearly a four-fold increase in the length of the tibia and femur over a period approximately 6 weeks, resulting in poor bone density and mineral content with reduced effective breaking strength [99]. In addition, the unnatural growth rate of leg bones, coupled with a large breast muscle (a high muscle-to-bone ratio) puts enormous compression on the immature skeleton and adds excessive torque and shear stress, causing bone deformities and mechanical trauma (e.g., microfracture and cleft formation) on the cartilaginous growth plates of the proximal leg bone, such as the tibial dyschondroplasia (osteochondrosis). Tibial dyschondroplasia is the most commonly observed non-infectious leg injury and is characterized by a mass of avascular cartilage in the metaphasis of the proximal ends of the tibiotarsus and tarsometatarsus [100,101]. In addition, mechanical trauma-induced chronic low-grade damage of the tibia and femur, a subclinical incidence of damage, provides a portal of entry for various bacteria, leading to bone inflammation and infection, such as bacterial chondronecrosis with osteomyelitis. Breeding-associated skeletal abnormalities could be the reason why a high level of leg weakness (lameness) has been found in modern meat-type chickens. As an external factor, heat stress causes bone weakness via multiple pathways, including (1) impaired GIT functions in food digestion and nutrient and mineral resorption, especially calcium resorption; (2) impaired local and systemic immunity, leading to the translocation of the pathogens and their metabolites to the leg bones and the brain through the bloodstream, causing bone damage and brain neuroinflammation; (3) impaired HPA functions, increasing CORT’s negative effects on bone remodeling, leading to increased apoptosis and necrosis of osteoblasts (reduced bone formation and increased bone resorption within a narrowed proliferative zone of the growth plate); and (4) impairment of the antioxidant system of bone cells, leading to oxidative stress [102,103,104].

Lameness (leg disorder) in broilers has been described as a locomotor system disorder and is one of the most serious welfare issues facing the modern broiler industry worldwide as it is highly prevalent and causes pain and suffering in billions of broilers [105,106]. Lameness encompasses a wide range of leg disorders in broilers with both infectious and non-infectious sources, such as tibial dyschondroplasia and bacterial chondronecrosis with osteomyelitis. Osteomyelitis is the most frequent form of infectious leg disease and is characterized by femoral or tibial head necrosis [107,108,109]. Effective prevention techniques for lameness are limited due to the multifactorial nature of the issue, influenced by many factors including growth rate, nutrition composition, genotype, environmental condition (e.g., ambient temperature and stocking density), and management practices [110,111]. Compassion in World Farming [112] reported that up to 96% of commercial broilers have some degree of musculoskeletal disorders. Heat stress and induced gut disorder (dysbiosis) could be critical external (environmental) factors causing lameness in broilers.

## 5. Heat Stress and Gut Microbiota

Gut microbiota is the collective name for the trillions of microorganisms living in an animal’s intestines, including more than 35,000 diversity species of known bacteria [113]. The function of gut microbiota is similar to a virtual endocrine organ, reacting to a variety of internal and external stimulations by producing various bioactivate factors to regulate food digestion, nutrient absorption, energy metabolism, immunity and antioxidative status, and hormonal and neurotransmitter release through the bidirectional communication of the microbiota–gut–brain axis [114,115,116,117]. A healthy gut microbiota is essential for the functions of the HPA and HPT axes [69,118,119,120]. Modulation of gut microbiota has become a novel strategy for improving hosts’ health and welfare under various conditions [121,122].

The mechanisms of the effects of heat stress on gut microbiota are not fully understood but could be related to stress directly and intricately caused by anatomical and functional disorders of the GIT. For example, blood flow in an animal under heat stress is reverted from organs such as the liver and intestines to dilated blood vessels of the peripheral tissue (skin) to facilitate heat loss. The poor circulation of the GIT decreases the proper amount of oxygen and nutrients supplied to intestinal epithelial cells, causing ischemia and hypoxia, leading to oxidative damage in the epithelial cells, impairing intestinal barrier integrity by disrupting tight junction proteins, consequently resulting in increased permeability to luminal endotoxin and pathogens (leaky gut) [123,124,125]. The released bacteria and bacterial products such as lipopolysaccharide (LPS) cause neuroinflammation, activating the HPA axis to release CORT (or cortisol in humans) and the automatic nervous system (the sympathetic and parasympathetic systems) to release epinephrine (EP) and norepinephrine (NE), leading to biological disorders in the animals. Heat stress also damages the composition of the gut microbiota in animals [126,127,128], i.e., decreasing beneficial bacteria but increasing pathogenic bacteria [129], resulting in interrupted gut homeostasis and triggered the release of proinflammatory cytokines and oxidative agents [130,131].

## 6. Probiotics and the Gut–Microbiota–Brain Axis in Broilers

### 6.1. The Potential Mechanisms of Probiotic Effects on Gut and Brain Functions

Probiotics (also called bio-friendly agents) are live microorganisms that improve the survival and implantation of beneficial microbes in the gut by: (1) altering the microbiota profile (bacterial diversity and/or population) with beneficial bacteria to prevent the growth of pathogens and the release of toxic metabolites by competing for the limited availability of nutrient and attachment sites; (2) producing bacteriocins (such as bacteriostatic and bactericidal substances) and short chain fatty acids (SCFAs) against pathogens to regulate the gut microenvironmental homeostasis; (3) activating intestinal digestive enzymes, increasing mineral solubility and nutrient absorption; (4) increasing the function of the endogenous antioxidant system and releasing heat shock proteins (HSPs) to reduce oxidative stress, inflammation, and acinar cell injury; (5) modulating the host’s local (gut) and systemic immune and related inflammatory responses to restore the intestinal barrier integrity, preventing pathogens from crossing the mucosal epithelial barrier; (6) stimulating the endocrine system and attenuating stress-induced disorders of the HPA and/or SMA axes via the microbiota–gut–brain axis; and/or (7) regulating the synthesis and secretion of neurotransmitters such as serotonin (5-HT) and tryptophan [132,133,134] (Figure 2). Using probiotics, as well as prebiotics and synbiotics, to target the gut microbiota with the aim of restoring the normal gut microbiota composition and intestinal homeostasis has become a biotherapeutic strategy for treating various diseases in humans and various animals, including poultry [135,136,137,138,139].

*Bacillus subtilis*, as one of the three most common species of probiotic products used in the United States [37,45], has been widely used in functional feed supplements, such as in several dairy and non-dairy fermented foods, for improving human health and well-being [46]. Similarly, *Bacillus subtilis* has been used as a probiotic with multiple functions in poultry [141]. As a growth promoter, for example, *Bacillus subtilis* has been demonstrated to improve chickens’ growth performance [142,143]; regulate intestinal microstructure [144] and digestive enzymes [145,146]; synthesize and release antimicrobial and antibiotic compounds [147,148,149,150]; increase immunity against gut inflammation and infectious diseases [151,152]; and enhance neurochemical activities [153,154] via the bidirectional communication between the gut and brain, regulating stress responses under various rearing conditions [155,156,157].

### 6.2. Bacillus subtilis and Broiler Skeletal Health in Broilers

Studies have elucidated how the gut microbiota and its metabolites influence bone metabolism via interactions with host metabolic, immune, and neuroendocrine systems through multiple pathways, including the microbiota–gut–brain axis, the microbiota–gut–immune axis, and the microbiota–gut–bone axis [5,54,55,158,159]. Alterations in microbiota homeostasis contribute to pathological bone loss and related diseases, such as osteoporosis (OP), in humans and various experimental animals by reducing nutrient and mineral resorption, increasing osteoclast proliferation, altering bone immune status, and/or changing the metabolisms of serotonin, corticosterone, and sex hormones [160,161,162]. In humans, OP characterized by low bone mass and deteriorated bone microarchitecture is a common skeletal disease. Primary OP is mostly seen in aged populations (nearly 22% of women and 7% of men who are older than 50), especially in postmenopausal women (estrogen deficiency) during the natural aging process. Secondary OP is a part of disease pathology, such as metabolic bone disorder seen in patients with inflammatory bowel diseases [163,164]. In both forms, one of the reasons for bone loss is an imbalance in the bone remodeling process, resulting from gut dysbiosis and related systemic inflammation [165,166,167]. In murine models, the manipulation of microbiota through the colonization of germ-free mice by the administration of antibiotics or stress hormones (such as glucocorticoids) significantly alters bone development, growth, remodeling, and strength, leading to bone loss and related bone damage [168,169,170]. The linkage between the microbiota and bone health has created several new scientific fields including osteomicrobiology [171,172] and ostroimmunology [173]. Probiotics, as well as fecal microbiota transplantation, prebiotics, and nutritional interventions, display functions in reversing gut microbiota dysbiosis, restoring gut permeability, and rescuing bone loss. Probiotic administration has been considered to be a potential novel target for regulating bone homeostasis to prevent bone diseases such as OP in humans, and in various animal models with induced bone loss [167,174,175,176]. However, studies have shown a range of variations in the incurred benefits of probiotics; and their efficiency is species-specific due to multiple factors, including the strains of the probiotics and the host’s physical and physiological state [53,54,55]. Our focus is on the effects of *Bacillus*-based probiotics on bone health in broilers; especially on the probiotic *B. subtilis*, as it has been used as an alternative to antibiotics in broiler production globally.

*Bacillus subtilis* has been documented to have broadly biochemical effects such as antimicrobial, anti-inflammatory, antioxidant, enzymatic, and immunomodulatory activities in humans and various animals [52,177]. *Bacillus subtilis* also releases several neuropeptides, such as human-like growth hormone [178,179], parathyroid hormone [180], and tryptophan [181,182,183]. All these biochemical compounds regulate bone development, especially tryptophan, the precursor of serotonin biosynthesis, which acts on the central serotonergic system due to the fact that it can pass the blood–brain barrier [184]. Central serotonin positively regulates bone development [185,186,187] (Figure 3).

Until recently, several *Bacillus*-based probiotics, especially probiotic *B. subtilis*, have been investigated in poultry for reducing or preventing breeding-associated lameness in broilers and OP in laying hens, as well as OP in humans and bone loss in several murine oral surgical models (Table 2).

In one of our studies (100), 1-day-old broilers were randomly assigned to two dietary treatments: a regular (basal) diet and a diet mixed with *B. subtilis* (1.0 × 10^6^ spores/g of feed) for a 44-day trail. Probiotic-fed broilers had greater bone mineralization, wall thickness, and size and weight of tibias and femurs, associated with higher serum calcium levels and a trend of lower levels of serum c-terminal telopeptide of type I collagen, a bone resorption indicator. Correspondingly, the levels of serotonin were increased in the Raphe nuclei, whereas the concentrations of norepinephrine and dopamine were decreased in the hypothalamus of probiotic-fed broilers. Similar results have been reported in another study, in which probiotic-fed broilers demonstrated a better latency to lie score, an indicator of leg muscle strength, with a greater tibial length, weight, and strength, as well as higher serum calcium and phosphorus concentrations [191]. Increases in the tibia’s physical traits (such as thickness of the medial and lateral wall, tibiotarsal index) and bone mineral levels (ash, calcium, phosphorus, bone mineral density, BMD, and bone mineral contents, BMC) were also found in broilers fed with a diet containing *B. subtilis* and *B. licheniformis* (BioPlus 28) [198]. Moreover, *Bacillus*-based probiotics increase bone health associated with suppression pathogens and the enhancement of beneficial bacteria. Broilers fed with a diet containing *B. subtilis* ATCC 6051a, increased tibia health associated with reduced cecal *Echerichia coli* and *Staphylococcus* spp. [192], reduced total Gram-negative bacteria but increased total lactic acid bacteria, with an increased gut absorption area (increased villus height, width, and area) [195,196]. Similar changes have been found in *B. subtilis*-fed laying hens [199] and *B. cereus*-fed Japanese quails [200]. Furthermore, *B. subtilis* C-3102 increased the hip bone mineral density in healthy postmenopausal women by reducing born resorption (decreased bone resorption makers: total urinary type I collagen cross-linked N-telopeptide and tartrate-resistant acid phosphatase) due to its regulation of the gut microbial balance, and increased and reduced the relative abundance of genus *Bifidobacterium* and *Fusibacterium*, respectively [189]. Both strains of *L**actobacillus* and *Bifidobacterium* have been used as probiotics for increasing bone health in humans and various animals [56,57,205]. These results indicate that dietary supplementation with the *B. subtilis-*based probiotic improves broiler bone health, most likely through increased intestinal nutrient and mineral absorption, modulated gut microbial balance, and reduced bone resorption by inhibiting sympathetic activity via the central serotonergic system. Brain serotonin simulates the development of bone mass through a reduction in sympathetic activity. This reduced sympathetic outflow in turn contributes to reduced sympathetic effects on bone resorption. Previous studies have reported that the activated sympathetic system negatively regulates bone mass by releasing norepinephrine, which binds to β2-adrenergic receptors expressed on osteoblasts and osteocytes [206,207]. The activated β2-adrenergic receptors on bone cells subsequently trigger a series of signaling pathways, leading to the inhibition of osteoblast proliferation (which has functions in bone formation) [208] and an increase in osteoclast formation (which has functions in bone resorption) [209,210]. Taken together, *B. subtilis-*based probiotics functionally prevent bone loss in broilers reared under thermoneutral conditions by recovering gut microbial homeostasis, restoring the balance between bone formation and bone resorption, and improving bone mineralization via the microbiota–gut–brain axis and the microbiota–gut–bone axis.

There is ample evidence supporting the notion that host gut microbial homeostasis is affected by various internal and external factors, and the effects of probiotics on host health are affected by the host’s pathophysiological characteristics in response to stress [124,211,212,213]. The use of probiotics to modulate gut microbiota may play an important role in preventing stress-induced gut inflammation and related bone damage [214]. The effects of probiotic *B. subtilis* and related strains on bone health have been examined in broilers exposed to various stressors, including heat stress. In one of our studies, probiotic *B. subtilis* (1 × 106 CFU/g feed) significantly increased bone mineral content, weight, size, and weight-to-length index, and reduced the robusticity index in both the tibia and femur of broilers exposed to hot ambient conditions (32 °C for 10 h daily from 15 to 42 days of age). Probiotic-fed broilers also had significantly lower serum concentrations of c-terminal telopeptide of type I collagen and heterophil-to-lymphocyte ratios, a stress indicator (the smaller the number, the less stress) [193,215]. In addition, probiotic-fed broilers also had significantly lower levels of serum tumor necrosis factor (TNF)-α, hepatic interleukin (IL)-6 and heat shock protein (HSP) 70 but higher levels of IL-10; however, peripheral serotonin and central serotonin, catecholamines (epinephrine, norepinephrine, and dopamine) and their metabolites were not affected by probiotic consumption. In addition, Cramer [216] reported that probiotic feeding improved breast muscle weight and alleviated oxidative deterioration of broilers undergoing heat stress. In another study, the administration of *B. subtilis* improved tibia traits, with higher skeletal muscle size, greater gut resorption area, and increased serum growth factor (GH) and insulin-like growth factor (IGF)-1 osberved in heat-stressed broilers [190]. The probiotic consumption also increased tibia mineralization (percentage of ash and calcium content) in broilers challenged with *Salmonella enteritidis* [197]. In a model of postmenopausal OP in ovariectomized mice, the probiotic consumption increased the bone mass and BMD and improved the bone microarchitecture associated with increased anti-inflammatory cytokine (IL-10) but reduced proinflammatory cytokines and restored the balance of Teg-Th17 cells [201]. Furthermore, in murine models of human oral diseases, probiotic supplementation increased osteoblasts in the maxilla of mice following mechanical loading (orthodontic tooth movement) [202] and reduced alveolar bone loss in the mandible and changes in the small intestines of rats with ligature-induced periodontitis [204]. The favorable effects on bone health were also seen in the model of periodontitis in rats fed with *B. subtilis* plus *B. licheniforis* [203]. In addition, *B. subtilis* has been used as an alternative to antibiotics in terms of improving intestinal health, such as increasing intestinal immunity and epithelial barrier integrity, in broilers infected with *Eimeria maxima* [217] or challenged with *Clostridium perfringens*-induced necrotic enteritis [49].

Many stressors, such as aging (associated with hormonal deficiency), high-fat diets (the typical “Westernized” diet of processed and fast foods), and environmental conditions (social stress, heat, or cold stress), affect a host’s microbiota homeostasis (dysbiosis) and disrupt intestinal integrity (leaky gut). This consequently exposes the host to immunosuppression, leading to chronic low-grade systemic inflammation and related distant organ dysfunction, such as devastating effects on bone remodeling [53]. Bone remodeling is a process of consistent interaction between osteoblasts and osteoclasts, with the dynamic equilibrium mostly being regulated by osteoclasts promoting proinflammatory cytokines such as IL-1β, IL-6, IL-17, TNF-α, interferon (IFN)-ϒ, and activator of nuclear factor kappa-B ligand (RANKL), and inhibiting anti-inflammatory cytokines such as IL-4 and IL-10 [188,201]. The RANKL/RANK/OPG pathway is one of the central regulators of bone remodeling; RANKL/RANK controls osteoclast formation and activation, whereas OPG (RANKL’s decoy receptor) functions as a bone-protective factor to prevent bone resorption. Proinflammatory cytokines (such as IL-1β and NTF-a) aid in bone resorption by promoting osteoclastognesis by increasing RANKL/RANK/OPG signaling, leading to OP. Probiotics, such as *B. subtilis* C-3102, reduce inflammatory factors and related RANKL expression in postmenopausal women [189]. The Treg–Th17 cell axis is another important pathway in the regulation of bone health via the gut–immune–bone axis [218]. Treg cells and Th17 cells are two subsets of CD4-positive T cells with opposite functions in regulating bone health. Treg cells have osteoprotective functions, suppressing bone resorption, whereas TH17 cells enhance osteoclastogenesis, increasing bone loss [201]. Th17 cells have been recognized as the major source of various osteoclastogenic cytokines such as TNF-α, RANKL, and IL-6, IL-17 [219]. A skewing of the Treg–Th17 cell equilibrium has been observed in *B. clausii* [201] and *Lactobacillus rhamnosus* [220]-treated postmenopausal OP mouse models by suppressing OVX-induced increasing bone marrow Th17 cells to inhibit the overstimulation of osteoclasts. Taken together, the potential mechanisms of *B. subtilis*’s improvement of skeletal health in heat-stressed broilers may be related to its functions in ameliorating heat-induced gut dysbiosis and related immune disorder-associated bone damage through regulation of the microbiota–immune–bone axis.

## 7. Conclusions and Perspectives

Heat stress has been recognized as a critical environmental stressor, negatively affecting the production, health, and welfare of farm animals, including poultry. In broilers, when body temperature increases past the thermoneutral zone it disturbs the physiological homeostasis, immunity, and production performance due to poor heat tolerance. Heat stress in particular damages the gut microstructure and the microbial composition and diversity, increasing intestinal permeability and systemic inflammation. Numerus studies, including some from our lab, have highlighted the role of gut permeability and triggered inflammatory pathways that disrupt bone integrity, resulting in bone disease such as osteoporosis in humans and osteoporotic bone damage in poultry. Probiotics consisting of *Bacillus* strains, such as *B. subtilis*, can modify gut microbial homeostasis to promote both intestinal and bone health via regulation of the microbiota–gut–brain axis and the microbiota–immune–bone axis. These data further suggest that probiotics may be a useful therapeutic strategy for improving health and production performance in poultry by decreasing stress-increased gut permeability and related bone damage such as lameness in broilers and osteoporosis in laying hens.

## Figures and Tables

**Figure 1 animals-11-01494-f001:**
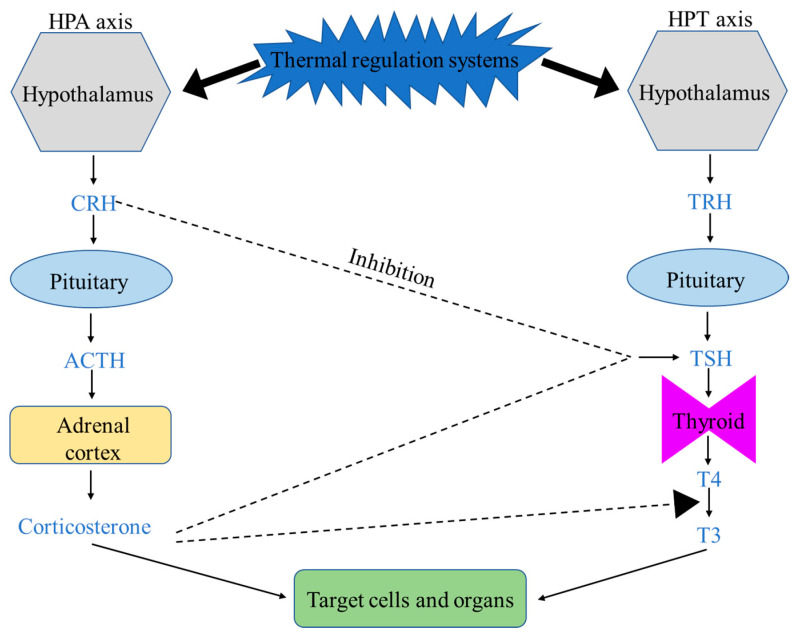
The major thermal regulation systems in chickens: the HPA axis and HPT axis and their interactions. ACTH: adrenocorticotropic hormone, CRH: corticotropin-releasing hormone, HPA: the hypothalamic–pituitary–adrenal, HPT: hypothalamic–pituitary–thyroid, T3: 3,5,3′-triiodothyronine, T4: thyroxine, TRH: thyrotropin-releasing hormone, TSH: thyroid-stimulating hormone.

**Figure 2 animals-11-01494-f002:**
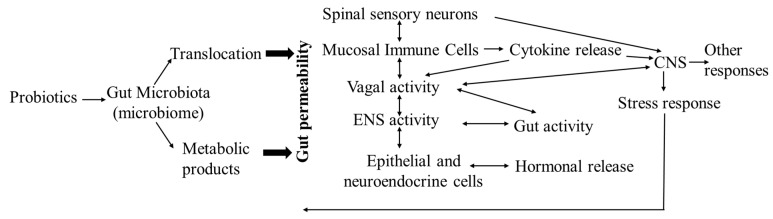
The effects of probiotics on the microbiota–host interaction at the level of the gastrointestinal mucosa via local neural, endocrine, and immune activities, influencing brain neurotransmitter expression, physiological homeostasis, and immunity, to regulate hosts’ stress responses (modified from Yarandi et al. [140]).

**Figure 3 animals-11-01494-f003:**
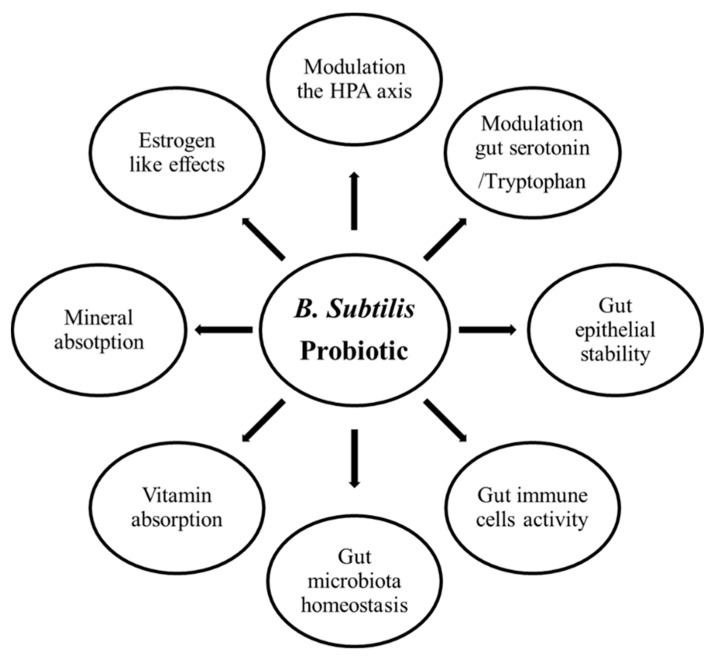
The schematic representation of the effects of *B. subtilis*-based probiotics on bone health via the microbiota–gut–brain axis, microbiota–gut–immune axis, and microbiota–gut–bone axis (modified from Amin et al. [188]).

**Table 1 animals-11-01494-t001:** Non-comprehensive list of commercially available *Bacillus*-based probiotics used in poultry production.

Trade Name	Bacillus Strain(s)	Manufacturer ^1^
Alterion^®^	*B. subtilis*	Adiddeo (Paris, France)-Novozyme(Copenhagen, Denmark). https://www.adisseo.com
B-ACT^®^	*B. licheniformis*	Agrihealth (Monaghan, Ireland) https://agrihealth.co.nz
Calsporin^®^	*B. subtilis* (C-3102)	Orffa.(Werkendam, Netherlands) https://orffa.com
Corrlink^TM^ ABS1781	*B. subtilis* (*B. velezensis* NRRL B-67259)	Elanco Animal Health (Fort Dodge, IA, USA) https://efsa.onlinelibrary.wiley.com
Clostat	*B. subtilis* PB6	Kemin Industries, Inc. (Des Moines, IA USA), https://www.kemin.com
Ecobiol^®^	*B. amyloliquefaciens* (CECT 5940)	Norel Animal Nutrition (Madrid, Spain) https://animal-nutrition.evonik.com
Enviva^®^ PRO 201 C	*B. subtilis*	Dupont-Danisco (Palo Alto, CA, USA) https://www.chemunique.co.za
Enviva^®^ PRO 202 GT	*B. amyloliquefaciens* (PTA-6507, NRRL B-50013, 50104)	Dupont-Danisco (Palo Alto, CA, USA) https://www.chemunique.co.za
FloraFix-BIRDS	*B. subtilis*	Biogrowcompany (Victoria, Australia) https://biogrowcompany.com/australia
Gallipro^®^ MS	*B. subtilis* (DSM5750), *B. licheniformis* (DSM5749)	Chr. Hansen, Inc. (Hoersholm, Denmark) https://www.chr-hansen.com
Gallipro^®^ Fit	*B. subtilis* (DSM 32324, 32325) *B. licheniformis* (DSM 25840)	Chr. Hansen, Inc. (Hoersholm, Denmark). https://www.chr-hansen.com
POULTRY-FEED	*B. subtilis, B. licheniformis*	Bionetix-International, Inc. (Quebec, Canada) www.bionextix-international.com
SPORULIN^®^	*B. subtilis*	Novus International, Inc. (St. Charles, MI, USA) http://www.novusint.com
Toyocerin	*B.cereus var toyoi*	Asahi Vet S.A. (Tokyo, Japan) https://trademark.trademarkia.com

Modified from the data presented by Grant et al. [50]; Markowiak and Slizewska [38]; and Ramlucken et al. [52]. ^1^ Accessed on 12 March 2021.

**Table 2 animals-11-01494-t002:** Effects of *Bacillus* probiotics on bone health.

Strain	Species (Model)	Bone Metabolism	Biochemistry/Biological Change	Reference
*B. subtilis* C-3102	Women	↑hip BMD, ↓bone resorption	↓uNTx, TRACP-5b, ↓genus *Fusobacterium*	[189]
*B. subtilis*	Broiler (HS)	↑tibia traits ^1^	↑production, absorptive epithelial area, ↑serum GH, IGF-1, and cholesterol, glucose	[190]
*B. subtilis*	Broiler	↑tibia traits	↑plasma Ca, P levels	[191]
*B. subtilis* ATCC 6051a	Broiler	↑Tibia P	↓cecal Escherichia coli, *Staphylococcus* spp.	[192]
*B. subtilis*	Broiler (HS)	↑tibia/femur BMC/traits	↓serum concentration of CTX, TNF-α	[193]
*B. subtilis* PB6	Broiler	↑tibia traits, tibia ash, Ca, P	↑serum OCN, BALP	[194]
*B. subtilis* *B. amyloliquefecieus*	Broiler	↑tibia breaking strength mineral	↑tibia breaking strength ↓TGNB, ↑TLAB	[195,196]
*B. subtilis*	Broiler (SE)	↑tibia ash and Ca	-	[197]
*B. subtilis* *B. licheniformis*	Broiler	↑tibia lateral and medial wall tibiotarsal index, tibia ash and P	-	[198]
*B. subtilis*	Laying hen	↑tibia traits	↑gut microbiota balance, egg traits ^2^	[199]
*B. cereus*	Quail	↑tibia traits	↑absorptive epithelial area	[200]
*B. Clausii*	Mouse (OVX)	↑bone mass, BMD, ↑bone microarchitecture	↑Treg cells↓Th17 cells, ↑anti-inflammatory cytokines, ↓proinflammatory cytokines	[201]
*B. subtilis*	Mouse (OTM)	↓osteoclasts,↑osteoblasts	-	[202]
*B. subtilis* *B. licheniformis*	Rat (periodontitis)	↓bone loss	↓inflammation	[203]
*B. subtilis*	Rat (periodontitis)	↓bone loss	↓inflammation	[204]

^1^ tibia trait: tibia weight, length, and strength and ^2^ egg trait: egg production, egg weight, egg mass, and eggshell quality: eggshell weight, eggshell thickness. ↑: increase; ↓: decrease, BALP: bone-special alkaline phosphatase; BMC: bone mineral content; BMD: bone mineral density; Ca: calcium; CTX: c-terminal telopeptide of type-I collagen (a marker of bone resorption); GH: growth hormone; HS: heat stress; IGF-1: insulin-like growth factor 1; OCN: osteocalcin; P: phosphorus; OTM: orthodontic tooth movement; OVX: ovariectomized (a postmenopausal osteoporotic animal model); SE: *Salmonella enteritidis*; TGNB: total Gram-negative bacteria; TLAB: total lactic acid bacteria; TNF-α: tumor necrosis factor-α; TRACP-5b: tartrate-resistant acid phosphatase isoform 5b (a marker of bone resorption); uNTx: urinary type I collagen cross-linked N-telopeptide (a marker of bone resorption).

## Data Availability

Data sharing not applicable.

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
