# Peer review of "Bacillus subtilis-Based Probiotic Improves Skeletal Health and Immunity in Broiler Chickens Exposed to Heat Stress"

_animals, 2021, doi:10.3390/ani11061494_

Round 1

Reviewer 1 Report

This is very interesting work but the paper is almost impossible to review without line numbers.  That being said, I'll do the best I can.

Simple summary - change "stress-associate" to "stress-associated"

Abstract - Change "Bacillus subtilis based" to "Bacillus subtilus-based"

Page 2 - 2nd paragraph - remove "the" between "to" and "temperature"

replace "is occurred" with "occurs"

Page 3 - 1st paragraph - insert 'abundant' between "most" and "hormone"

Page 3 - 3rd paragraph - replace "amount" with "amounts"

Page 3 - 5th paragraph - remove "the" between "Especially" and "modern"

Page 5 - 3rd paragraph - replace "likes" with "is similar to" between "microbiota" and "a"

Page 5 - 4th paragraph - insert "to" between "related" and "stress"

Insert "by" between "caused" and "anatomical"

delete "the most" between "from" and "organs"

Page 6 - 3rd paragraph - replace "was" with "has been" between "health" and "widely"

Page 7 - delete "were" between "care' and "followed"

replace "day 43" with "43 d"

replace "great" with "greater" between "had" and "bone

replace "day 43" with "43 d"

add "s" to "regulate"

add "s" to "broiler"

Page 8 - remove "the" from the title of Table 1

Page 9 -  remove "the" from the title of Table 2

Page 10 - delete "were"

change "heats" to "heat"

remove "the" between "that" and "dietary" 

Page 11 - put Eimeri maxima in italics

put Clostridium perfringens in italics. You have Clostridium misspelled

Page 12 - change "negative" to "negatively"

insert "it" between "zone" and "distributes"

Author Response

REVIEWER 1

This is very interesting work but the paper is almost impossible to review without line numbers. That being said, I'll do the best I can.

Thanks for the reviewer’s comments and suggestions. The followings have been changed at the relative locations marked with yellow highlighted.

Simple summary - change "stress-associate" to "stress-associated"

Abstract - Change "Bacillus subtilis based" to "Bacillus subtilus-based"

Page 2 - 2nd paragraph - remove "the" between "to" and "temperature"; replace "is occurred" with "occurs"

Page 3 - 1st paragraph - insert 'abundant' between "most" and "hormone"

Page 3 - 3rd paragraph - replace "amount" with "amounts"

Page 3 - 5th paragraph - remove "the" between "Especially" and "modern"

Page 5 - 3rd paragraph - replace "likes" with "is similar to" between "microbiota" and "a"

Page 5 - 4th paragraph - insert "to" between "related" and "stress"; Insert "by" between "caused" and "anatomical"; delete "the most" between "from" and "organs"

Page 6 - 3rd paragraph - replace "was" with "has been" between "health" and "widely"

Page 7 - delete "were" between "care' and "followed" ; replace "day 43" with "43 d"; replace "great" with "greater" between "had" and "bone; replace "day 43" with "43 d"; add "s" to "regulate"; add "s" to "broiler"

Page 8 - remove "the" from the title of Table 1 2

Page 9 - remove "the" from the title of Table 2

Page 10 - delete "were" ; change "heats" to "heat"; remove "the" between "that" and "dietary"

Page 11 - put Eimeri maxima in italics, put Clostridium perfringens in italics. You have Clostridium misspelled

Page 12 - change "negative" to "negatively"; insert "it" between "zone" and "distributes"

Reviewer 2 Report

The paper that I reviewed undertakes the topic of heat stress in chickens and various effects of Bacillus subtilis probiotic, with a special emphasis on gut-brain axis. I am very curious, why the Authors focused only on B. subtilis? I think that the first part of the paper is excellent. Problem starts with chapter 3 and Bacillus subtilis part. It seems that those two parts have been written by different Authors, which is not a bad thing, but the style should be more harmonized. Now it makes impression of two papers randomly put together.

I think the Authors should work on the “probiotic” part of the paper, I recommend that the number of probiotics included in this paper is extended so that the paper is more general and thus interesting to broader audience. Also, I do not fully understand why the Authors cited the whole tables with the results (each table containing data from ONE paper)? In the review papers, the tables usually synthesize data from multiple sources. And I would recommend the Authors to do the same here.

Last but not least, Introduction is a must and a review paper does not end with Conclusions (see below).

Specific comments:

Suggestion to the title: “(..) Bacillus subtilis probiotic improves (…)”

I suggest to include some short introduction in the paper, that will show the context, motivation and goals of this review paper. Please justify focusing solely on Bacillus subtilis probiotic.

The manuscript does not include line numbers, therefore in my comments I refer to the page/paragraph number.

Figure 1 and the text describing Figure 1 on Page 2 should have some references added (there isn’t a single one)

Page 3, First paragraph – “The most hormone produced” (adjective missing); “to decease or increase” should read “to decrease or increase”?

Page 5, 4. Heat stress and microbiota: please rephrase: “The function of gut microbiota likes a virtual endocrine organ”

Tables 1-5. I completely do not understand including five papers from two research papers. In the review paper, I would rather recommend using some systemic review of multiple papers on a given topic rather than pasting data from two random papers. I think that publishing already published records without any added value does not make much sense for this type of publication. I recommend the Authors to broaden their scope, and rebuild the tables to include more summarized data from different studies (but without citing the entire datasets).

I STRONGLY recommend re-writing the following subchapters:

3.2. Bacillus subtilis and broiler skeletal health under thermoneutral condition

3.2. Bacillus subtilis and broiler skeletal health under heat stress

So that, the tables include the overview of the summarized effects of bacillus subtilis (rather than numerical results), so that it is easier to follow the various effects in different studies. Also, I would rather see the text to have a better structure, that would guide the reader through the various effects of the probiotic supplementation in a more reader-friendly way. At the moment it is a little bit too technical compared to the first part of the review

Conclusion is a form typical to a research paper. In review paper, a Summary is typically used.  

Author Response

Thanks the reviewer’s comments and suggestions, and the manuscript has been revised accordingly.

-, The paper that I reviewed undertakes the topic of heat stress in chickens and various effects of Bacillus subtilis probiotic, with a special emphasis on gut-brain axis. I am very curious, why the Authors focused only on B. subtilis? I think that the first part of the paper is excellent. Problem starts with chapter 3 and Bacillus subtilis part. It seems that those two parts have been written by different Authors, which is not a bad thing, but the style should be more harmonized. Now it makes impression of two papers randomly put together.

The reasons of this manuscript focused on Bacillus subtilis are: 1) Genus Bacillus (B.) is paraphyletic to Lactobacillates (including Lactobacillus and streptococcus); 2) B. subtilis and Lactobacillus, as well as Enterococcus faecium, are the three most common bacterial species used in commercial probiotic products for animal production (Markowiak and Slizewska, 2018; Joerger and Ganguly, 2017) including poultry (Heak et al., 2018; Jia et al., 2020; Khan et al., 2020); 3) the effects of probiotic strains of Lactobacillus and Bifidobacterium on bone health in humans and rodents have been comprehensively  reviewed (Kiousi et al., 2019; Schepper et al., 2017); 4) probiotic Bacillus spp. (such as B. subtilis, B. pumilus, B. polyfermenticus, B. licheniformis, B. cerue, B. coagulans, B. clauisii, B. amyloliquefaciens) used as human probiotics (Lee et al., 2019) and as growth promoters or antibiotic alternatives to enhance growth and health in poultry have recently been reviewed (Grant et al., 2018; Abd El-Hack et al., 2020) but their functions in regulating bone development and health have not been well understood; and 5) gut microbiota and its metabolites affect bone homeostasis through the gut-bone axis, however, probiotic effects are spices-special (Behera et al., 2020; Li et al., 2021; Zaiss et al., 2019). In addition, B. subtilis has been documented to have broadly biochemical effects such as antimicrobial, anti-inflammatory, antioxidant, enzymatic, and immunomodulatory activities in humans and various animals (Ramlucken et al., 2020; Ruiz Sella et al., 2021). Bacillus subtilis also releases several neuropeptides, such as human-like growth hormone (Ozdamar et al., 2009; Sahin et al., 2015), parathyroid hormone (Karimi et al., 2018), and tryptophan (Shasaltaneh et al., 2013; McAdams and Gollnick, 2014; Bjerre et al., 2017). All these biochemical compounds regulate bone development, especially tryptophan. Plasma concentrations of tryptophan, as the precursor of serotonin biosynthesis, acts on the central serotonergic system due to it can pass the brain-blood-barrier (Gao et al., 2017). Central serotonin positively regulates bone development (Driessier and Baldock, 2010; Ducy, 2011; Lavoie et al., 2017). The aim of our studies as well as several others is to examine if B. subtilis increases bone health in poultry, by which it prevents or reduces lameness in broilers and osteoporosis in laying hens.  The information has been provided in relative sections.    

References

Abd El-Hack ME, El-Saadony MT, Shafi ME, Qattan SYA, Batiha GE, Khafaga AF, Abdel-Moneim AE, and Alagawany M. 2020. Probiotics in poultry feed: A comprehensive review. J Anim Physiol Anim Nutr (Berl). 104(6):1835-1850.

Behera J, Ison J, Tyagi SC, and Tyagi N. The role of gut microbiota in bone homeostasis. Bone. 135:115317.

Bjerre K, Cantor MD, Nørgaard JV, Poulsen HD, Blaabjerg K, Canibe N, Jensen BB, Stuer-Lauridsen B, Nielsen and B, Derkx PM. 2017. Development of Bacillus subtilis mutants to produce tryptophan in pigs. Biotechnol Lett. 39(2):289-295.

Driessler F, and Baldock PA. Hypothalamic regulation of bone. J Mol Endocrinol. 2010 Oct;45(4):175-81. doi: 10.1677/JME-10-0015. Epub 2010 Jul 26. PMID: 20660619.

Ducy P. 2011. 5-HT and bone biology. Curr Opin Pharmacol. 11(1):34-8.

Gao K, Mu CL, Farzi A, Zhu WY. 2020. Tryptophan Metabolism: A Link between the Gut Microbiota and Brain. Adv Nutr. 11(3):709-723.

Grant A, Gay CG, and Lillehoj HS. 2018. Bacillus spp. as direct-fed microbial antibiotic alternatives to enhance growth, immunity, and gut health in poultry. Avian Pathol. 47(4):339-351.

Heak C, Sukon P, and Sornplang P. 2018. Effect of direct-fed microbials on culturable gut microbiotas in broiler chickens: a meta-analysis of controlled trials. Asian-Australas J Anim Sci. 31(11):1781-1794.

Jha R, Das R, Oak S, and Mishra P. 2020. Probiotics (Direct-Fed Microbials) in Poultry Nutrition and Their Effects on Nutrient Utilization, Growth and Laying Performance, and Gut Health: A Systematic Review. Animals (Basel). 10(10):1863.

Joerger RD and Gamguly A. 2017. Current Status of the Preharvest Application of Pro- and Prebiotics to Farm Animals to Enhance the Microbial Safety of Animal Products. Microbiol Spectr. 5(1).

Jones RM, Mulle JG, and Pacifici R. Osteomicrobiology: The influence of gut microbiota on bone in health and disease. Bone. 115:59-67.

Karimi M, Behzadian F, Rouhaninejad H, and Yari S. 2018. A Feasibility Study to Evaluate Bacillus subtilis as a Host for Producing Recombinant Human Parathyroid Hormone. Avicenna J Med Biotechnol. 10(3):147-151.

Khan S, Moore RJ, Stanley D, and Chousalkar KK. 2020. The Gut Microbiota of Laying Hens and Its Manipulation with Prebiotics and Probiotics To Enhance Gut Health and Food Safety. Appl Environ Microbiol. 86(13):e00600-20.

Kiousi DE, Karapetsas A, Karolidou K, Panayiotidis MI, Pappa A, and Galanis A. 2019. Probiotics in Extraintestinal Diseases: Current Trends and New Directions. Nutrients. 11(4):788.

Lavoie B, Lian JB, and Mawe GM. 2017. Regulation of Bone Metabolism by Serotonin. Adv Exp Med Biol. 1033:35-46.

Lee NK, Kim WS, and Paik HD. 2019. Bacillus strains as human probiotics: characterization, safety, microbiome, and probiotic carrier. Food Sci Biotechnol. 28(5):1297-1305.

Li J, Ho WTP, Liu C, Chow SK, Ip M, Yu J, Wong HS, Cheung WH, Sung JJY, and Wong RMY. 2021. The role of gut microbiota in bone homeostasis. Bone Joint Res. 10(1):51-59.

Markowiak P. and Śliżewska K. 2018. The role of probiotics, prebiotics and synbiotics in animal nutrition. Gut Pathog. 10:21.

McAdams NM, and Gollnick P. 2014. The Bacillus subtilis TRAP protein can induce transcription termination in the leader region of the tryptophan biosynthetic (trp) operon independent of the trp attenuator RNA. PLoS One. 9(2):e88097.

Ozdamar TH, Sentürk B, Yilmaz OD, Calik G, Celik E, and Calik P. 2009. Expression system for recombinant human growth hormone production from Bacillus subtilis. Biotechnol Prog. 25(1):75-84.

Ramlucken U, Roets Y, Ramchuran SO, Moonsamy G, van Rensburg CJ, Thantsha MS, and Lalloo R. 2020. Isolation, selection and evaluation of Bacillus spp. as potential multi-mode probiotics for poultry. J Gen Appl Microbiol. 66(4):228-238.

Ruiz Sella SRB, Bueno T, de Oliveira AAB, Karp SG, and Soccol CR. 2021. Bacillus subtilis natto as a potential probiotic in animal nutrition. Crit Rev Biotechnol. 41(3):355-369.

Åžahin B, Öztürk S, Çalık P, and Özdamar TH. 2015. Feeding strategy design for recombinant human growth hormone production by Bacillus subtilis. Bioprocess Biosyst Eng. 38(10):1855-65.

Schepper JD, Irwin R, Kang J, Dagenais K, Lemon T, Shinouskis A, Parameswaran N, and McCabe LR. 2017. Probiotics in Gut-Bone Signaling. Adv Exp Med Biol. 1033:225-247.

Shasaltaneh MD, Moosavi-Nejad Z, Gharavi S, and Fooladi J. 2013. Cane molasses as a source of precursors in the bioproduction of tryptophan by Bacillus subtilis. Iran J Microbiol. 5(3):285-92.

Zaiss MM, Jones RM, Schett G, and Pacifici R. 2019. The gut-bone axis: how bacterial metabolites bridge the distance. J Clin Invest. 129(8):3018-3028.

-, I think the Authors should work on the “probiotic” part of the paper, I recommend that the number of probiotics included in this paper is extended so that the paper is more general and thus interesting to broader audience. Also, I do not fully understand why the Authors cited the whole tables with the results (each table containing data from ONE paper)? In the review papers, the tables usually synthesize data from multiple sources. And I would recommend the Authors to do the same here.

It has been rewritten as suggested.

-, Last but not least, Introduction is a must and a review paper does not end with Conclusions (see below).

Introduction has been added and conclusion has been rewritten as a summary.

Specific comments:

-, Suggestion to the title: “(..) Bacillus subtilis probiotic improves (…)”

Agree with the reviewer. The title has been changed as suggested. It is “Bacillus subtilis-based probiotic improves skeletal health and immunity in broiler chickens exposed to heat stress”, so it is much closely related to the aim of the manuscripyt.

-, I suggest to include some short introduction in the paper, that will show the context, motivation and goals of this review paper. Please justify focusing solely on Bacillus subtilis probiotic.

            Agree with the reviewer. The introduction has been added.

-The manuscript does not include line numbers, therefore in my comments I refer to the page/paragraph number.

-, Figure 1 and the text describing Figure 1 on Page 2 should have some references added (there isn’t a single one)

            The references have been cited.

-, Page 3, First paragraph – “The most hormone produced” (adjective missing); “to decease or increase” should read “to decrease or increase”?

            It has been changed based on the suggestion received from another reviewer.

-, Page 5, 4. Heat stress and microbiota: please rephrase: “The function of gut microbiota likes a virtual endocrine organ”

            It has been changed based on the suggestion received from another reviewer.

-, Tables 1-5. I completely do not understand including five papers from two research papers. In the review paper, I would rather recommend using some systemic review of multiple papers on a given topic rather than pasting data from two random papers. I think that publishing already published records without any added value does not make much sense 2 for this type of publication. I recommend the Authors to broaden their scope, and rebuild the tables to include more summarized data from different studies (but without citing the entire datasets).

Agree with the reviewer. The tables have been rewritten, with much more information presented.

-, I STRONGLY recommend re-writing the following subchapters:

3.2. Bacillus subtilis and broiler skeletal health under thermoneutral condition

3.2. Bacillus subtilis and broiler skeletal health under heat stress

So that, the tables include the overview of the summarized effects of bacillus subtilis (rather than numerical results), so that it is easier to follow the various effects in different studies. Also, I would rather see the text to have a better structure, that would guide the reader through the various effects of the probiotic supplementation in a more reader-friendly way. At the moment it is a little bit too technical compared to the first part of the review

            Agree with the reviewer. These two subchapters have been rewritten with the modified tables.

-, Conclusion is a form typical to a research paper. In review paper, a Summary is typically used.

      It has been changed to summary based on the suggestion.